# Adverse Drug Reactions (ADRs) of Montelukast in Children

**DOI:** 10.3390/children9111783

**Published:** 2022-11-21

**Authors:** Abdullah Al-Shamrani, Saleh Alharbi, Sumayyah Kobeisy, Suzan A. AlKhater, Haleimah Alalkami, Turki Alahmadi, Aisha Almutairi, Adel S. Alharbi, Abdullah A. Yousef

**Affiliations:** 1Department of Pediatrics, Prince Sultan Military Medical City, AL Faisal University, P.O. Box 7897, Riyadh 11159, Saudi Arabia; 2Dr. Soliman Fakeeh Hospital, P.O. Box 2537, Jeddah 21461, Saudi Arabia; 3Department of Paediatrics, Umm Al-Qura University, Mecca 24382, Saudi Arabia; 4College of Medicine, Imam Abdulrahman Bin Faisal University, Dammam 34212, Saudi Arabia; 5Department of Pediatrics, King Fahd Hospital of the University, Al-Khobar 31952, Saudi Arabia; 6Department of Pediatrics, Abha Maternity & Children Hospital, P.O. Box 62521, Abha 1650, Saudi Arabia; 7Department of Pediatrics, Faculty of Medicine, King Abdulaziz University, Jeddah 21589, Saudi Arabia; 8Department of Pediatrics, College of Medicine, Qassim University, P.O. Box 7897, Buraidah 51452, Saudi Arabia

**Keywords:** montelukast, singular, neuropsychiatry, side effect, adverse effect

## Abstract

Montelukast is a leukotriene receptor antagonist (LTRA) commonly prescribed for asthma, allergic rhinitis and sleep-related breathing disorders. Recently, some studies have reported several adverse events, such as neuropsychiatric disorders and sleep disturbances, among children. Objective: To obtain more insight into the safety profile of montelukast for children with asthma, allergic rhinitis and sleep-related breathing disorders. Method and results: We retrospectively studied all adverse drug reactions to montelukast among 385 children 6 months or older in six tertiary centers over a two-year period. A total of 89.6% were asthmatic, 50% had allergic rhinitis and 13.6% had sleep-related breathing disorders; Singulair was the most common type of montelukast used (67.9%). This study reported a high prevalence of adverse drug reactions among 123 patients (31.9%), predominantly in those aged 4–9 years (52.8%), followed by adolescents (24.4%) and toddlers (22.8%). Two (ADRs) were reported in 9.8% of the children, while three or more were reported in 5.5%. Sleep disturbance was the most common (ADRs), affecting 15.1% of participants (overlap was common; 5.5% of children experienced sleep difficulties, 4.4% experienced sleep interruption and decreased sleep, and 1.82% experienced nightmares), followed by agitation (10.4%), pain (9.4%) and hyperactivity (6.8%). No serious (ADRs) were reported. Eleven percent of families faced difficulties in purchasing montelukast, and only 57% of families had insurance. Misconceptions were common (9.8% reported it to be a steroid, while 30.6% believed it to be a bronchodilator). Although 81% of the families believed it was an effective and preventive medication, 5.3% stopped the drug due to concern about side effects, especially agitation (3%) and nightmares (0.6%). Conclusion: These data demonstrate that montelukast is effective, but the associated adverse neuropsychiatric drug reactions are more prevalent than those reported in the literature. In particular, sleep disturbance, agitation, pain and hyperactivity were observed. Pediatricians should be aware of such (ADRs). Misconceptions about montelukast are still common, and parental counseling and urgent epidemiological studies are needed to quantify the risk for management plans.

## 1. Introduction

Montelukast is a leukotriene receptor antagonist (LTRA) that is one of the most common medications used for asthma and other medical conditions. LTRAs function by inhibiting inflammatory mediators of bronchoconstriction, and they are prescribed primarily as adjuvant medication to inhaled corticosteroids for patients with step 3 or higher asthma, although they might be prescribed as an alternative to inhaled corticosteroids for mild asthma [1]. In 2009, the US Food and Drug Administration (FDA) warned clinicians about the use of montelukast, which included certain important observations that may involve neuropsychiatric changes [2]. Due to the high prevalence of asthma among children and the serious side effects of montelukast, we decided to investigate whether this popular medication, which is commonly used under limited conditions for pediatric patients, is associated with any neuropsychiatric event in children in six main cities in Saudi Arabia [3,4,5,6]. Asthma is a common childhood condition, and its prevalence has increased in the last two decades from 8 to 23% [7,8]. In the Kingdom of Saudi Arabia, the highest prevalence is reported in Hafoof (33%), and the lowest is reported in the southern region of the kingdom (Abha) (7%) [9]. Unfortunately, the majority of asthma cases in the Kingdom of Saudi Arabia are uncontrolled, as reported by Dr. Jahdali et al. using Asthma Control Test (ACT) [10]. A similar study was conducted by Dr. Aslan et al., which showed that 50% of asthma cases among children were uncontrolled in a tertiary center in Riyadh [11]. Asthma is an often heterogeneous disease with a wide range of presentations from mild cough to severe exacerbation with different asthma phenotypes [12,13]. Asthma cases are often divided into two main groups: a younger age group and an older age group. Such classification helps health care providers diagnose and manage the patients’ conditions. In patients older than 5 years of age, it is easy to diagnose asthma, as it is straightforward based on clinical presentation. Such patients respond well to asthma therapy, including bronchodilators or corticosteroids. There are different types of montelukast that are used for this group as add-ons to inhaled steroids, and they are often chewable tablets (4, 5 or 10 mg) [12,14,15]. However, it is a real challenge for health care providers to diagnose asthma in patients younger than 5 years, as patients of this age are unable to undergo spirometry may have similar conditions that share the same symptoms of cough, wheezing and shortness of breath.

Currently, the majority of asthma cases worldwide are still uncontrolled, even in Saudi Arabia, per different reports; fortunately, there has been some improvement according to recent publications [16,17,18,19]. Montelukast plays an important role in the management of asthma, nasal allergies and sleep-related breathing disturbances, and the most common types of montelukast are 4 mg granules or chewable tablets [14]. Allergic rhinitis and sinusitis are frequent causes of nocturnal coughing and are most often misdiagnosed as asthma. They share triggers with asthma, and there are two main types of presentation: Type 1 (non-inflammatory /watery) is the predominant type, and the child presents with clear nasal discharge, sneezing and nasal itching; and Type 2 (inflammatory) is the less common type, and the child presents with a blocked nose and signs of nasal obstruction [20,21,22]. Commonly used medications include intranasal corticosteroids and oral antihistamines [20]. Intranasal antihistamines may be necessary to improve the treatment of nonallergic rhinitis with eosinophilia (NARES) or vasomotor rhinitis [21,23]. Longitudinal studies have confirmed that both allergic rhinitis and positive allergic skin tests are risk factors for asthma [24]. Montelukast is often used as an adjuvant to inhaled steroids for asthma with proven efficacy [25], and allergic rhinitis is often associated with asthma and called united airway disease [26]. Chronic rhinosinusitis with and without nasal polyps may aggravate some symptoms, particularly coughing, which may be attributed to severe asthma [27,28]. Sleep-related breathing disorders are characterized by prolonged partial upper airway obstruction and/or intermittent complete obstructive apnea that disrupts normal ventilation during sleep and normal sleep patterns [29]. Classification based on polysomnography results as proposed by Dayyat, E. et al. stated that it is essential to differentiate obstructive sleep apnea (OSA) from other disorders as the treatment and complications are different [30]. The prevalence is 1–5% among children, with the peak prevalence occurring between the ages of 2–8 years and the peak of symptoms often occurring in the middle of the second year, which could be related to the peak growth of lymphoid tissue [31,32,33]. While the prevalence of habitual snoring is estimated to be approximately 15%, although it has been reported to be as high as 30% in the pediatric age group, the ratio of the prevalence of habitual snoring and OSA varies from 4:1 to 6:1 [34]. OSA occurs equally among boys and girls during the prepubertal stage [32,35,36,37,38]. OSA has been reported to cause significant school problems among children, such as short attention span, aggressive behavior, poor academic performance, excessive daytime sleepiness, behavioral disorders, and multiple other problems, including cardiac growth and metabolic consequences [39,40,41]. Inhaled steroids and montelukast are effective medical therapies for mild forms of OSA [13,42,43,44,45,46,47].

This study was intended to evaluate all potential side effects that have been reported or addressed by searching the MEDLINE database and focusing on neuropsychiatric attributes after starting montelukast, such as sleep disturbance, anxiety, oppositional, depression and any sort of suicidal attempt [48,49]. During this decade, numerous studies were released discussing the neuropsychiatric effects of leukotrienes, but the relationship was not direct and remained controversial [50,51].

## 2. Materials and Methods

We retrospectively evaluated all potential adverse drug reactions (ADRs) to montelukast among children aged 6 m–17 years. This study was conducted in six tertiary centers in the Kingdom of Saudi Arabia [Prince Sultan Military Medical City (PSMMC) in Riyadh, Qassim University, Imam Abdulrahman Bin Faisal University in Dammam, Dr. Suleiman Fakeeh Hospital, King Abdul-Aziz University Hospital in Jeddah and Maternity and Children Hospital in Abha, Saudi Arabia] among children with a physician-confirmed diagnosis of asthma, allergic rhinitis or OSA until May 2022. The sample size was calculated using Epitools (https://epitools.ausvet.com.au/oneproportion accessed on 1 September 2022). A sample of 385 children treated with montelukast was the minimum sample to have 80% power to ascertain a prevalence of adverse effects with a 95% confidence interval and a precision of 5%. An electronic questionnaire was used for the all patients, the focus was on children with symptoms suggestive of asthma, nasal allergies, or sleep-related breathing disorders. The use of montelukast alone or as adjunct therapy to inhaled corticosteroids and/or long-acting bronchial agonists currently or in the past was also recorded. Children with known central nervous system (CNS) disorders, associated comorbidities, or those who did not receive montelukast were excluded. The occurrence of montelukast-related adverse drug effects was assessed by questioning using three approaches. First, parents were asked about the three potential indications for montelukast, the types of montelukast and potential adverse effects and their knowledge of montelukast. However, for those children with sleep-related breathing disorders, we double checked if the symptoms worsened after being prescribed montelukast, as the disease itself can cause similar adverse effects to montelukast. The statistical analysis was performed using SPSS for Windows, version 21.0 (SPSS Inc., Chicago, IL, USA). The data are reported as frequencies and proportions. A nonparametric test was employed for variables with nonnormal distribution. Independent sample t tests were used to compare data between participants of different ages. Chi-square tests were used to compare categorical groups. *p* < 0.05 with a 95% confidence interval was considered statistically significant.

## 3. Result

A total of 385 patients were included in this study (boys, 64.4% and girls, 35.6%). Toddlers aged 1–3 years, children aged 4–9 years and adolescents aged over 10 years were the most affected (30.8%, 44.3%, and 23.4%, respectively). Boys were the most affected (62% asthma, 66% allergic rhinitis and 71% sleep-related breathing disorders). The most common indication for prescribing montelukast was asthma, with 89.6% (46.6% isolated and 43% associated), followed by allergic rhinitis at 50% (7% isolated) and asthma and allergic rhinitis at 28.1%; the least common indication was sleep-related breathing disorders alone at 13.6% (0.8% isolated) (Table 1). Singulair was the most common type of montelukast (67.9%), followed by Airfast (20.2%), and 9.3% of the parents did not know the brand name (Table 2). The most common concentration prescribed was 4 mg powder (33.9%), followed by 4 mg chewable tablets (31.9%) and 5 mg chewable tablets (26.7%); 1.8% patients used the 10 mg tablets, and 5.7% did not know the concentration. Families were satisfied with montelukast (78% agreed that it is a good preventive medication, 86% preferred to use it, and 81.1% believed that it is an effective medication). Only 5.3% stopped montelukast due to concerns about side effects. A total of 31.9% of the patients reported at least one adverse effect. The most common was observed among school children aged 4–9 years (52.8%), followed by adolescent children (24.4%) and toddlers (22.8%). The distribution of ADRs was as follows (Table 3): sleep problems (15.1%), agitation (10.4%), pain (9.3%), hyperactivity (6.8%), short attention span (3.1%), aggression (2.1%) and headache (1.82%). There were different types of sleep problems, including sleep difficulties (5.5%), decreased sleep (4.4%), interrupted sleep (4.4%) and nightmares (1.82%). A total of 9.8% of patients reported two complaints, and 5.5% reported three or more complaints. Twenty percent of the parents were worried about the side effects of montelukast; furthermore, 11% faced difficulties in purchasing the medication, and only 57% had insurance. Regarding misconceptions, 9.8% believed montelukast was a steroid, while 30.6% believed it was a bronchodilator. The duration of montelukast usage ranged from one month to 108 months, with a mean of 8 months; the youngest patient was 6 months old, and the oldest was 17 years old. Regarding pain, predominantly abdominal pain (6.32%), chest and muscle pain (2.1%), and hyperactivity with short attention span or agitation (3.6%) were reported. No serious adverse effects, required hospitalizations or deaths were reported in this study. Recovery was reported for most patients when they discontinued the medication.

## 4. Discussion

Montelukast is a cysteinyl leukotriene receptor (D4 and E4) antagonist that is commonly used as prophylactic treatment for asthma and in the treatment of allergic rhinitis and sleep-related breathing disorders [1]. This study reported a high prevalence of adverse drug reactions among 123 patients (31.9%), sleep disturbance was the most common (ADRs). Since 2008, there has been an increasing number of reports on potential adverse drug reactions to montelukast, including irritability, aggressiveness, and sleep disturbances, usually occurring after 1 week of commencing therapy and occurring in up to 10% of children [3,4,52,53,54,55,56,57,58]. The World Health Organization (WHO) pharmacovigilance database reported multiple adverse effects, including depression, headache, aggression, suicidal ideation, anxiety, insomnia, abnormal behavior, nightmares, shortness of breath, abdominal pain, nausea, rash, dizziness, myalgia and muscle spasm [53]. The FDA advised caution when montelukast and other LTRAs are prescribed due to the potential adverse neuropsychiatric events (such as depression and suicidality) as a precaution [50,53,58]. On the other hand, a large nested case—control study with 1920 asthmatic patients matched for age, sex, and geographic region did not detect a significant positive association between montelukast and neuropsychiatric events among children [54].

However, a more recent nested case—control study reported an odds ratio of nearly 2 for new-onset neuropsychiatric events in children prescribed montelukast in the year before the event [43].

A retrospective study of reported (ADRs) in the Netherlands and worldwide data from the WHO indicated that nightmares were frequently reported after commencing montelukast in the first week of treatment [53], with a reported odds ratio of 78.04 (95% CI 70.0–87.1) among children. Glockler-Lauf et al. reported that (ADRs) occur within a week of starting montelukast, and children exposed to this drug tend to have a quick recovery of sleep disturbance upon cessation [54]. Given the high reported incidences of such (ADRs) among both adults and children in the USA, Canada, and Europe [3,4,43,53,54,57,59,60,61,62,63,64]. Table 3 summarizes reported adverse effects of montelukast among children.

It is essential to know the prevalence and pattern of the (ADRs) of montelukast among children with asthma, allergic rhinitis and OSA in Saudi Arabia. In this study, we found that almost one-third of the patients reported (ADRs) that could be serious enough to disturb their lifestyle. This report stresses the high prevalence (15.1%) of sleep disturbance among children using montelukast, including sleep difficulties (5.5%), interrupted sleep (4.4%), insomnia (4.4%), and nightmares (2.8%); three patients reported increased sleep duration (0.8%), and 5.5% reported more than three adverse sleep complaints. Previous studies have reported different types of sleep disturbance and depression effects associated with montelukast [4,43,49,51,52,64]. Nightmares were the main sleep-related cause to discontinue montelukast [64]. This report highlights different sleep disorders compared to what has been reported in the literature.

Neuropsychiatric events were predominant, and the causative mechanism for montelukast is still not known [54]. Agitation (10.4%) was the single predominant adverse neuropsychiatric effect, which was slightly different from the aggression previously reported by Harman et al. [53] or anxiety as reported by Glockler-Lauf et al. [43].

The prevalence of pain (9.3%) was higher than that in a previous report (abdominal pain, 6.3%, chest and muscle pain, 2.1%); however, this high number could be exaggerated due to functional abdominal pain and growing pain, which are common in this age group [51,63].

Hyperactivity was reported in 6.8% of participants, while attention deficit was reported in 1.8%, which is similar to previous studies [4,49,51,59]. While Po-Yu Huang et al. reported that montelukast Montelukast does not increase the risk of attention-deficit/hyperactivity disorder in pediatric asthma patients [59]. Hyperactivity or attention deficit hyperactivity disorder (ADHD) could be the presentation of sleep disturbance. Since a minority of the patients experienced sleep disturbance, it is difficult to reach a conclusion [30,35,41].

Headaches were experienced by 1.82% of participants and were reported most frequently in the Dutch database (ROR, 2.26; 95% CI: 1.61–3.19) [53]. Fatigue was reported in 1.5% of participants, and other adverse effects were reported in less than 1% (skin rash, vomiting, visual disturbance, itching and learning difficulties). No participants in this study showed any suicidal behavior, and no deaths occurred, as was cautioned by the US FDA in 2008 [5].

Approximately 80% of the parents believed that montelukast is a good preventive and effective medication; however, the Pearson’s chi-square result was 4.455. Our result is similar to that in a Dutch study conducted in 2016, in which few families were aware of the side effects of montelukast, although this study emphasizes the limited knowledge of parents about this medication [65]. Eleven percent of participants’ families faced difficulties in purchasing the medication, and only 57% had insurance. Misconceptions were common (9.8% believed it was a steroid, while 30.6% believed it was a bronchodilator) [66,67]. The duration of usage ranged from one month to 108 months, with a mean of 8 months; the youngest patient was 6 months old, and the oldest was 17 years old. This study emphasizes the importance of explaining the potential adverse effects to parents when prescribing montelukast for children. Risk management plans should be explained carefully. Pediatric health care providers should be aware of the risk of neuropsychiatric events and sleep disturbance associated with montelukast use and should advise the patient to observe, monitor and report any potential ADRs). The limitation of the study, is the un expected high prevalence of adverse drug reactions than reported in literatures, relatively small sample size for such common disease, urgent epidemiological studies are strongly recommended.

## 5. Conclusions

This article offers a comprehensive overview of the safety of montelukast in clinical practice.

This study confirms the concern of a high prevalence of adverse neuropsychiatric effects among patients receiving montelukast, especially agitation, sleep disturbance, pain, and hyperactivity. Families and physicians should be aware of the (ADRs) of montelukast when it is prescribed; fortunately, no serious side effects have been reported. Misconceptions about montelukast are still common; therefore, education of the families is necessary. Further epidemiological studies are urgently needed to quantify the risk for management.

## Figures and Tables

**Table 1 children-09-01783-t001:** Prevalence of diseases in Montelukast patients.

Disease	Number	Percent
Sleep disorder of breathing	3	0.8%
Allergic rhinitis (AR)	27	7.0%
Allergic rhinitis and sleep disorder of breathing	10	2.6%
Asthma	180	46.6%
Asthma, sleep disorder of breathing	10	2.6%
Asthma and AR	108	28.1%
asthma, AR, Sleep disorder of breathing	29	7.6%
Asthma, AR, and other	18	4.7%
	385	100

**Table 2 children-09-01783-t002:** Brand of montelukast.

		Frequency	Percent	Valid Percent	Cumulative Percent
Valid	Airfast	78	20.2	20.3	20.3
	Don’t know	36	9.3	9.4	29.6
	Montel	9	2.3	2.3	31.9
	Singulair	262	67.9	68.1	100.0
	Total	385	99.7	100.0	
Missing	−999	1	0.3		
Total		386	100.0		

**Table 3 children-09-01783-t003:** (ADRs) of montelukast in children.

Montelukast Side Effect	Number	Percent
1—Agitation	40	10.4%
2—Sleep problems	58	15.1%
(a) Decrease sleep (b) Interrupted sleep (c) Increase sleep (d) Sleep difficulties (e) Night mares (f) Decrease and interrupted sleep (g) Decrease and difficult sleep (h) ≥Interrupted sleep	17 17 3 21 7 11 7 21	4.4% 4.4% 0.8% 5.5% 1.82% 2.9% 1.82% 5.5%
3—(a) Hyperactivity (b) Hyperactivity and low attention (c) Hyperactivity and agitation	26 7 7	6.8% 1.82% 1.82%
4—Pain (a) Abdominal pain (b) Chest pain (c) Muscle pain (d) Chest and abdomen	36 24 8 8 5	9.35% 6.23% 2.1% 2.1% 1.3%
5—Low attention	12	3.1%
6—Aggression	8	2.1%
7—Headache	7	1.82%
8—Fatigue	6	1.5%
9—Skin rash	4	1.0%
10—Vomiting	3	0.8%
11—Itching	3	0.8%
12—Visual disturbance	3	0.8%
13—Learning difficulty	1	0.3%

## Data Availability

Ethics approval was obtained from the IRB Prince Sultan Medical city. (IRB-2020-0416). The approval date is 20 December 2020.

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
