# Peer review of "Adverse Drug Reactions (ADRs) of Montelukast in Children"

_children, 2022, doi:10.3390/children9111783_

Round 1

Reviewer 1 Report

Thank you for providing me the opportunity to review this manuscript. In this paper, Al-Shamrani et al. conducted a retrospective study that evaluated the safety profile of the use of montelukast among children with asthma, allergic rhinitis and sleep-related breathing disorders.

The paper needs a revision.

Firstly, authors should use only one definition for Adverse Drug Reactions (ADRs) throughout the entire manuscript.

As for each section separately, kindly find below my comments.

Abstract

The abstract is readable and describes the results precisely.

In line 29, please remove “adolescent children” and replace it with “adolescents”.

Introduction

The introduction section needs a revision. Asthma, allergic rhinitis, chronic rhinosinusitis and sleep-related breathing disorders are extensively analyzed. Authors should focus on the drug, its use on dealing with such diseases, and the mondelukast-induced ADRs according to studies’ findings.

Please define the “OSA” abbreviation.

The aim of the study should be moved to the last paragraph of the introduction section.

Materials and Methods

There is a deviation between the numbers of tertiary centers, where the study was conducted, that were reported to the Materials and Methods section compared with those reported to the Introduction. Please make the appropriate correction.

Please determine the number of patients who were evaluated with an electronic questionnaire and with a direct one.

Please define the “CNS” abbreviation.

Results

The description of the main characteristics of study’s participants in a table would make the results section more readable and clear.

Please define the abbreviation mentioned to Table 1.

Please remodel Table 2, as the data of the columns do not correspond to the lines’ ones and to those that were mentioned in the manuscript.

Author Response

Dear respected reviewer in children journal 

 Good day to you 

I hope this mail find you well 

It is a great pleasure to us to such edit, please have a look  for our response hoping to satisfy you

Looking forward  to hear from you

Best regards

Dr. Shamrani 

Primary author

10-11-2022

Reviewer 2 Report

The authors retrospectively reviewed this article on the adverse effects in children who used montelukast. I have some comments about this article.

1. The introduction is quite long. The introduction and discussion sections contain common sentences. The introduction should be shortened and rearranged in accordance with the purpose.

2. There are shifts in the rows in Table 2. Table 3 is not required, I suggest reconsidering.

3. "These data demonstrate that montelukast is effective.." the sentence has been written. You have no objective data to confirm this statement. I recommend you review it.

4. It would be appropriate to refer to this recently published article on this topic as well.

"Relationship between montelukast and behavioral problems in preschool children with asthma. Allergol Immunopathol (Madr). 2022"

Author Response

Dear respected reviewer 

 Good day to you

 Hope this mail find you well 

 Thanks for revising and editing the paper, great comment really

 please, have a looks to the response letter,  wish to meet your standards.

please, let us know if that's fine to you 

 Many thanks 

Shamrani 

Primary author 

Round 2

Reviewer 1 Report

Thank you for improving the manuscript according to the submitted suggestions.

There are some more improvements in the Discussion and Refererences section that I mentioned in my first review.

The discussion section should initiate with an abstract of your main findings and subsequently with the interpretation and comparison of your results with other previous studies'. Any further information is superfluous and has no place in the discussion section. Moreover, the third paragraph which represents the main content of the discussion is huge and it should be divided into smaller ones, in order to be more readable. 

No limitations of the study were mentioned in the discussion section.

Authors should modify the reference list according to journal's instructions.

Author Response

many thanks  for the first reviewer for his/her great input 

 very constructive and excellent feed back, reflect how expert in the research  especially,  modifying  the discussion part 

 hope the current  change meet the standard for the reviewer and journal 

 best regards 

 dr shamrani

Reviewer 2 Report

The authors performed the necessary adjustments. That's enough. 

Author Response

many thanks   for the great support 

dr shmarni